

# Diversity, host-specificity and stability of sponge-associated fungal communities of co-occurring sponges

Mary T.H.D. Nguyen and Torsten Thomas

Centre for Marine Bio-Innovation and School of Biological, Earth and Environmental Sciences, University of New South Wales, Sydney, NSW, Australia

## ABSTRACT

Fungi play a critical role in a range of ecosystems; however, their interactions and functions in marine hosts, and particular sponges, is poorly understood. Here we assess the fungal community composition of three co-occurring sponges (*Cymbastela concentrica, Scopalina* sp., *Tedania anhelans*) and the surrounding seawater over two time points to help elucidate host-specificity, stability and potential core members, which may shed light into the ecological function of fungi in sponges. The results showed that ITS-amplicon-based community profiling likely provides a more realistic assessment of fungal diversity in sponges than cultivation-dependent approaches. The sponges studied here were found to contain phylogenetically diverse fungi (eight fungal classes were observed), including members of the family *Togniniaceae* and the genus *Acrostalagmus*, that have so far not been reported to be cultured from sponges. Fungal communities within any given sponge species were found to be highly variable compared to bacterial communities, and influenced in structure by the community of the surrounding seawater, especially considering temporal variation. Nevertheless, the sponge species studied here contained a few "variable/core" fungi that appeared in multiple biological replicates and were enriched in their relative abundance compared to seawater communities. These fungi were the same or highly similar to fungal species detected in sponges around the world, which suggests a prevalence of horizontal transmission where selectivity and enrichment of some fungi occur for those that can survive and/or exploit the sponge environment. Our current sparse knowledge about sponge-associated fungi thus indicate that fungal communities may perhaps not play as an important ecological role in the sponge holobiont compared to bacterial or archaeal symbionts.

## INTRODUCTION

Fungi are ecologically important in terrestrial environments performing vital functions as free-living decomposers, nutrient cyclers, parasites, commensals, and mutualists (*Webster & Weber, 2007*). The global fungal richness has been estimated between 1.5 and 1.6 million species (*Hawksworth, 1991*, *2001*). Most of our current understanding of the ecology and function of fungi is derived from studies of cultured fungal isolates, mostly from

Corresponding author
Torsten Thomas,
t.thomas@unsw.edu.au

terrestrial environments (*Richards et al., 2012*). In comparison, little is known about the diversity and ecology of fungi in the marine environment as it is estimated that only ~0.6% of all cultured and studied fungi are marine derived (*Burgaud et al., 2009*; *Kis-Papo, 2005*). Marine fungi form an ecological, but not a taxonomic group and can be classified, according to *Kohlmeyer & Volkmann-Kohlmeyer (1990)*, as "obligate marine" (i.e., those that grow and sporulate exclusively in marine habitats) and "facultative marine" (i.e., those that are from freshwater and terrestrial milieus, but are able to grow and possibly sporulate in marine environments). As more studies emerge, marine fungi are increasingly considered to play an important ecological role as saprotrophs, parasites, commensals or mutualists in marine ecosystems (*Hyde et al., 1998*), and have been reported to be associated with higher marine organisms, such as macroalgae, corals and sponges (*Richards et al., 2012*).

Sponges are sessile, filter-feeding organisms that play a crucial role in benthic communities through nutrient cycling, including carbon, silicon, oxygen and nitrogen, and the provision of habitats for a range of fauna and flora (*Bell, 2008*). Sponges can harbour microbial communities from all three domains of life, archaea, bacteria and eukaryotes (fungi, microalgae, and protozoa), which can account for up to 40–60% of the sponge's volume in some species (*Hentschel, Usher & Taylor, 2006*). The diversity of sponge-associated bacteria and archaea has been extensively studied showing often the presences of stable, sponge-specific symbiont communities (*Thomas et al., 2016*; *Hentschel et al., 2002*, *2003*; *Webster & Taylor, 2012*). In contrast, eukaryotic symbionts have received little attention and even less is known about their function and interaction with the sponge host (*Taylor et al., 2007*; *Webster & Taylor, 2012*).

To date, 22 orders of Ascomycota and eight orders of Basidiomycota have been found in sponges (*Yu et al., 2013*), largely through culture-dependent approaches (*Morrison-Gardiner, 2002*; *Wang, Li & Zhu, 2008*; *Li & Wang, 2009*), which aimed at the discovery of biologically active, secondary metabolites (*Taylor et al., 2007*; *Baker et al., 2009*; *He et al., 2014*; *Höller et al., 2000*; *Liu et al., 2010*). Ubiquitous fungal genera (e.g., *Aspergillus*, *Penicillium*, *Trichoderma*, and *Acremonium*) have been isolated from multiple sponge species worldwide suggesting these may be either "generalist" and/or are preferentially detected due to their ease of cultivation (*Richards et al., 2012*). In fact, cultivation conditions as well as sample preparation methods have likely biased the comparison between studies (*Anderson, Campbell & Prosser, 2003*). To overcome these issues, molecular techniques that amplify and sequence the 18S rRNA gene or the internal transcribed spacer (ITS) of the ribosomal RNA gene operon directly from DNA extracted from samples have been applied to examine fungal diversity in a range of environments (*De Beeck et al., 2014*; *Döring et al., 2000*; *Gardes & Bruns, 1993*; *Anderson, Campbell & Prosser, 2003*). PCR-primer bias towards certain fungal groups or non-target DNA, however, are drawbacks of this approach (*Wang et al., 2014*). Culture-dependent and culture-independent approaches have however been shown to capture different parts of fungal communities in deep-sea sediments (*Singh et al., 2012*; *Jebaraj et al., 2010*) and soil (*Jeewon & Hyde, 2007*).

Few studies have considered the diversity, ecological function or the nature of sponge-fungi interactions. Recent studies on fungal diversity in sponges present conflicting reports on their host-specificity. For example, *Gao et al. (2008)* reported distinct fungal communities between two co-occurring Hawaiian sponges and also to the surrounding seawater using denaturing gradient gel electrophoresis (DGGE)-based ITS analysis. *He et al. (2014)* found that the fungal communities differed on the order level between some Antarctic sponge species and *Rodríguez-Marconi et al. (2015)* further suggested that there is a high degree of host specificity of fungi in Antarctic sponges. However, it should be noted that both studies lacked biological replications. In contrast, *Naim, Smidt & Sipkema (2017)* recently reported low host-specificity and suggested the presence of fungi in sponges to be rather "accidental".

Aside from these community-wide studies, there has been some observational evidence for sponge-fungi interactions. This includes the vertical transmission of an endosymbiotic yeast in the marine sponge *Chondrilla* sp. (*Maldonado et al., 2005*) and the putative horizontal gene transfer of a fungal mitochondrial intron into the genome of the sponge *Tetilla* sp. (*Rot et al., 2006*). In addition, the sponge *Suberites domuncula* has been suggested to recognize fungi via the D-glucans on their surfaces (*Perović-Ottstadt et al., 2004*) and ascomycetes of the genus *Koralionastes* have been reported to have a unique physical association with crustaceous sponges (*Kohlmeyer & Volkmann-Kohlmeyer, 1990*). And finally, the ability of many sponge-derived fungi to produce bioactive compounds has been suggested to contribute to host defense (*Höller et al., 2000*; *Imhoff, 2016*; *Proksch et al., 2003*, *2008*; *Wiese et al., 2011*; *Yu et al., 2013*). These studies indicate that certain close symbiotic interactions between fungi and sponges exist; however, this conclusion seems to be not necessarily supported by culture-dependent and -independent community-wide analyses.

The aim of the current study is to assess the diversity of the fungal community of sponges using cultivation-based and cultivation-independent ITS community profiling to examine their suitability to describe sponge-associated fungal diversity. Fungal communities of three co-occurring sponges and the surrounding seawater were assessed over two time points to help elucidate host-specificity, stability and potential core members, which may shed light into the ecological function of fungi in sponges.

# MATERIALS AND METHODS

## Sample collection

Sponges were sampled at Bare Island in Botany Bay, NSW, Australia (33°59′S, 151°14′E) on two separate occasions, on the 13 November 2014 and on the 4 May 2016. At each sampling event, seawater samples (SW) and three specimens of *Cymbastela concentrica* (C), *Scopalina* sp. (S) and *Tedania anhelans* (T) sponges were collected by SCUBA diving at a depth of 7–10 m and within an area of about 20 × 20 m. Sampling of sponges was performed under the scientific collection permit P13/0007-1.1 issued by the New South Wales Department of Primary Industries. Sponge specimens were identified by the morphological characteristics and their locality as per our previous study (*Fan et al., 2012*). Samples were placed individually into Ziploc® bags with seawater and then transported in

buckets filled with seawater to the laboratory at ambient temperature (travel time approximately 30 min). Sponges samples were processed immediately for cultivation or frozen at −80 °C for subsequent DNA extraction. Seawater (200 mL) were vacuum filtered onto 0.22 µm filters (Whatman, Sigma-Aldrich, St. Louis, MO, USA) in replicates and used immediately for cultivation or frozen at −80 °C for subsequent DNA extraction.

## Cultivation of fungi

Sponge tissues were processed as described by *Wang, Li & Zhu (2008)*. Briefly, sponge samples were washed three times in sterile calcium/magnesium-free seawater (CMFSW; 25 g NaCl, 0.8 g KCl, 1 g $Na_2SO_4$, and 0.04 g $NaHCO_3$ per 1 L) to remove natural seawater from the sponge and the outer surfaces of the samples were sterilized with 70% ethanol. Two different cultivation methods were applied: (a) sponges were sliced into thin sections (approximately 1 $cm^2$ and 1–2 mm thick) and placed directly onto agar plates (listed below) and (b) sponge tissue was homogenized at maximum speed for 50–60 s with a dispersing homogenizer (Ulta-Turrax TR50; IKA, Selangor, Malaysia). The homogenate was diluted with sterile seawater ($10^0$, $10^{-1}$, $10^{-2}$) and 100 µL of each dilution was plated. For SW two methods were applied: (a) 200 mL of seawater were vacuumed filtered onto 0.22 µm Whatman filters (Sigma-Aldrich, St. Louis, MO, USA) and filters were directly placed onto agar plates and (b) 1 mL of seawater was directly plated onto agar plates. Triplicates of each sample and preparation method were plated onto three different media: (1) peptone yeast glucose agar (PYG; 1.0 g glucose, 0.1 g yeast extract, 0.5 g peptone and 15 g Difco-bacto agar per 1 L, pH 8.0); (2) Dextrose potato agar (BD Difco™, North Ryde, Australia) and (3) Gause I (20 g starch, 1.0 g $KNO_3$, 0.5 g $K_2HPO_4$, 0.5 g $MgSO_4 \cdot 7H_2O$, 0.5 g NaCl, 0.01 g $FeSO_4$ and 15 g Difco-bacto agar per 1 L). All media was made up with 0.22 µm filtered and autoclaved seawater from Bare Island. The plates were incubated at 18–20 °C for 1–2 months until fungal growth was visible. Every isolate was picked and transferred onto new PYG, Potato dextrose and Gause I agar plates. The resulting pure cultures were stored in sterile artificial seawater (ASW; 23.38 g NaCl, 2.41 g $MgSO_4$, 1.19 g $MgCl_2$, 1.47 g $CaCl_2.2H_2O$, 0.75 g FCl, and 0.17 g $NaHCO_3$ per 1 L) in 2 mL cryogenic vials (Sigma-Aldrich, St. Louis, MO, USA) at 4 °C.

## Identification of fungal isolates

DNA extraction from fungal isolates were conducted using the CTAB method (*Lee, Milgroom & Taylor, 1988*) with modifications. Briefly, fungal mycelia were added to 1 mL of CTAB buffer, 3–6 1 mm glass beads (Sigma-Aldrich, St. Louis, MO, USA) and 10 µL of mercaptoethanol and bead beaten (TissueLyser II; Qiagen, Hilden, Germany) at maximum speed for 8 min. Samples were heated at 65 °C for 10 min, then extracted with phenol:chloroform:isoamyl alcohol (25:24:1) and DNA was precipitated with isopropanol. DNA was dissolved in 50 µL of pure water and used for PCR amplification. PCR was conducted in 25 µL reactions consisting of 12.5 µL of Econotaq master mix, 9.5 µL of water, 1 µL of each forward primer ITS1f-F (10 µM) (5′ TTGGTCATTTAGAGGAAGTAA 3′) and reverse ITS4 (5′ TCCTCCGCTTATTGATATGC 3′) (*White et al., 1990*) and 1 µL of template DNA. PCR conditions were 95 °C/2 min, then 94 °C/30 s, 53 °C/30 s, 72 °C/45 s

for 35 cycles and 72 °C/5 min. Amplicon products were assessed by gel electrophoresis, cleaned with Exosap-IT (ThermoFisher Scientific, Waltham, MA, USA) and sequenced with the BigDye Terminator v3.1 chemistry (Applied Biosystems, Austin, TX, USA) and an Applied Biosystems 3730 DNA Analyzer at the Ramaciotti Centre for Genomics (University of New South Wales, Sydney, NSW, Australia). Sequences were manually quality trimmed using Sequences Scanner v1.0 software and forward and reverse sequencing reads were assembled (when applicable) using BioEdit v7.2.5 (Ibis Biosciences, Carlsbad, CA, USA).

## Sequence analysis of ITS amplicon community profiling

Total DNA were extracted from sponge samples and seawater filters from the two sampling time points using the Power Soil DNA Isolation kit (Qiagen, Hilden, Germany) following the manufacturer's instructions. Fungal ITS amplicon sequencing was conducted by Molecular Research LP (Mr. DNA, Houston, TX, USA) using the IST1f-ITS4 primers. PCRs were conducted with a HotStarTaq Plus Master Mix kit (Qiagen, Hilden, Germany) under the following conditions: 94 °C for 3 min, followed by 28 cycles of 94 °C for 30 s, 53 °C for 40 s and 72 °C for 1 min, and a final elongation step at 72 °C for 5 min. After amplification, PCR products are checked in 2% agarose gel to determine the success of amplification and the relative intensity of bands. PCR samples were pooled together in equal proportions based on their molecular weight and DNA concentrations and purified using calibrated Ampure XP beads. The pooled and purified PCR products were then used to prepare a Illumina DNA library. Sequencing was performed on the Illumina MiSeq sequencing platform and 2 × 300 bp chemistry following the manufacturer's guidelines. Because of the variable length of the ITS region (400–800 bp), forward and reverse sequences could often not be assembled into contigs. In addition, reverse reads had generally lower quality than the forward reads and therefore only the forward ITS amplicon sequence reads (300 bp) were quality filtered and analyzed together with the isolate sequences from above. All sequences were quality filtered with a maximum expected error threshold of 1 and minimum length of 250 bp and then clustered into operational taxonomic units (OTUs) at 97% similarity using the UPARSE pipeline (Usearch v9.2) (*Edgar, 2013*). Chimeras were removed using UCHIME (*Edgar et al., 2011*) and taxon classification of OTUs were conducted using the UNITE ITS reference database (*Abarenkov et al., 2010*) with the utax command in the Usearch program. Lowest taxon classification at a confidence level of 70% or above were considered and checked with the Basic Local Alignment Search tool (BLASTn) against the non-redundant nucleotide database from the National Centre for Biotechnology Information (NCBI). Non-fungal sequences were removed from further analysis. The commands for the entire ITS amplicon analysis are available in the Supplemental Information. Raw ITS-amplicon sequences are available through the NCBI Sequence Read Archive under Bioproject ID: PRJNA419577, accession number SRP125576.

## Sequence analysis of 16S rRNA community profiling

Given the high variability of fungal communities observed between biological replicates (see below), we wanted to understand if this is a peculiar aspect of the samples we took and how we processed them or if this is due to real biological variation. We therefore
also analyzed all samples for the bacterial community composition, which has been shown to be very consistent between replicates of the three sponges analyzed here (*Fan et al., 2012*; *Esteves et al., 2016*). Bacterial 16S rRNA gene amplicon sequencing of sponge and seawater samples were therefore conducted using primers 515F (5′-GTG CCA GCM GCC GCG GTA A-3′) and 806R (5′-GGA CTA CHV GGG TWT CTA AT-3′) with the Illumina MiSeq sequencing platform and 2 × 250 bp chemistry at the Ramaciotti Centre for Genomics (University of New South Wales, Sydney, NSW, Australia), according to the methodology described by *Caporaso et al. (2012)*. Bacterial 16S rRNA sequences were processed using the MiSeq SOP pipeline (Mothur v1.37.3) (*Schloss et al., 2009*). Briefly, raw forward and reverse sequence reads were assembled into contigs, quality filtered and aligned to the SILVA 16S rRNA gene reference alignment v102 (*Quast et al., 2012*). Sequences were filtered to only include overlapping regions, pre-clustered to merge all sequences within three mismatches (difference = 3) and checked for chimeras using the UCHIME algorithm (*Edgar et al., 2011*). To separate chloroplasts from cyanobacteria, sequences were first classified using the SILVA reference v119 (*Quast et al., 2012*) with a 60% confidence threshold and sequences classified as chloroplasts were removed. The rest of the sequences were re-classified using the RDP training set release 9 (*Cole et al., 2013*) and sequences classified as "unknown" or "mitochondria" were removed before clustering into OTUs at 97% similarity. OTU matrix was then sub-sampled to the size of the smallest sample (4,193 sequences). The commands for this analysis are available in the Supplemental Information.

## Statistical analysis

The fungal OTU matrix of sponge and seawater samples were used to calculate the species richness estimate (Chao1) and Shannon's index using the summary.single command in Mothur v1.39.0. Fungal community coverage was estimated using Good's coverage (= 1-(number of singleton OTUs/number of reads)). Comparison of beta-diversity was conducted with permutational multivariate analysis of variance (PERMANOVA) (*Anderson, 2001*) of Bray–Curtis dissimilarities of relative abundance and presence-absence values at the OTU level. Variability of communities was analyzed using Multivariate Homogeneity of Group Dispersions (PERMDISP) (*Anderson, Ellingsen & Mcardle, 2006*; *Anderson, 2006*) based on Bray–Curtis dissimilarities. Heatmap and statistical analysis were performed in the R statistical program language (*R Core Team, 2014*) using the vegan package (*Oksanen et al., 2010*). Scripts are provided in the Supplemental Information.

## RESULTS

### Diversity of fungal communities through culture-dependent and independent methods

Culture-dependent and -independent approaches were applied to sponge and seawater samples collected in 2014. Cultivation yielded a total of 108 isolates and after redundant sequences were removed, resulted in 42 unique isolate sequences (see Table S1 for details), which clustered into eight OTUs at 97% similarity. In contrast, 26 OTUs were obtained

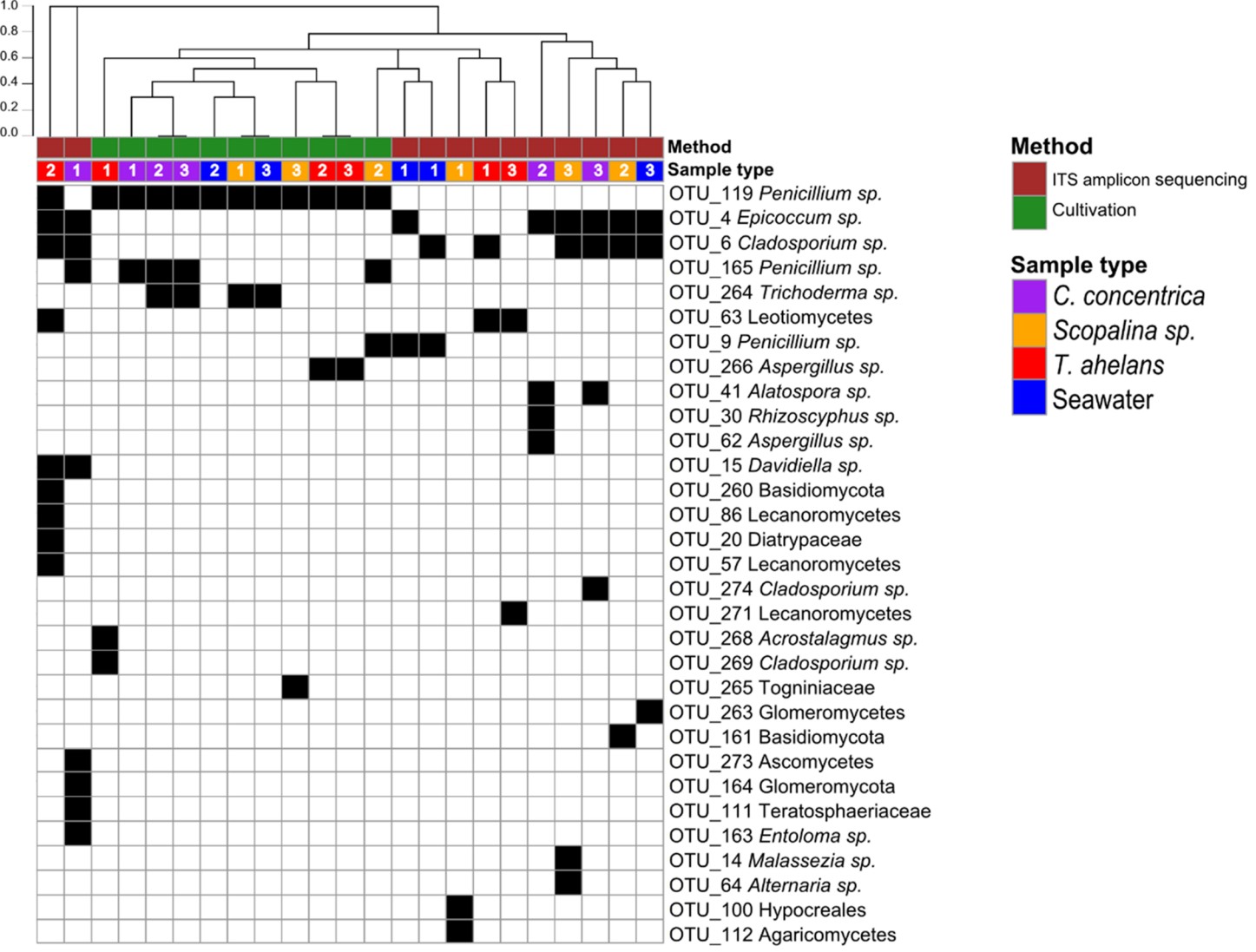

**Figure 1** **Presence (black)-absence (white) map of the fungal OTUs obtained from ITS-amplicon sequencing (brown) and cultivation (green) for samples (biological triplicates) collected in 2014.** Columns were clustered based on Bray–Curtis dissimilarity using hierarchical clustering with the "average" method (scale depicts the percentage of dissimilarity). Samples are indicated in purple: *C. concentrica*, orange: *Scopalina* sp., blue: seawater, and red: *T. anhelans*. Numbers 1, 2, and 3 indicate sample replicates.

via ITS-amplicon sequencing clustered at 97% similarity. Cultivation obtained one fungal division, Ascomycota, and four fungal orders (Eurotiales, Hypocreales, Diaporthales, and Capnodiales) compared to ITS-amplicon sequencing, which obtained three divisions (Ascomycota, Basidiomycota, and Glomeromycota) and 12 orders (Pleosporale, Capnodiales, Eurotiales, Leotiales, Malasseziales, Xylariales, Helotiales, Lecanorales, Teloschistales, Hypocreales, Agaricales, and Diaporthales) (Fig. 1). Significant differences (PERMANOVA *P*-value <0.008) in the fungal community compositions were observed overall and for each sponge species and seawater between the cultivation-dependent and -independent assessments (Fig. 1), with the cultivation method producing a lower fungal diversity (Shannon's index of 0.77 ± 0.32) compared to the ITS amplicon

sequencing (1.17 ± 0.54; one-way ANOVA, *P*-value = 0.04). Although a higher fungal diversity was obtained by ITS-amplicon-based community profiling, the cultivation method yielded five unique OTUs not present in the ITS amplicon data (Fig. 1). OTU_266 (*Aspergillus* sp.), OTU_286 (*Acrostalagmus* sp.) and OTU_269 (*Cladosporium* sp.) were only cultivated from *T. anhelans*, OTU_265 (family *Togniniaceae*, genus unclassified) was cultivated from *Scopalina* sp., and OTU_264 (*Trichoderma* sp.) was cultivated from *C. concentrica*, *Scopalina* sp. and seawater. Three OTUs assigned to the genus *Penicillium* were found to overlap between the two methods (OTU_119, OTU_165, and OTU_9). OTU_119 was commonly cultivated from all sample types and found once in *T. anhelans* ITS-amplicon sequencing. The most common OTUs detected in the ITS amplicon data were OTU_4 (*Epicoccum* sp.) and OTU_6 (*Cladosporium* sp.) and were observed in all sample types (i.e., three sponge species and seawater) (Fig. 1).

## Analysis of the temporal stability and host-specificity of sponge-associated fungal communities

Since cultivation recovered only a smaller proportion of the total fungal diversity found compared to ITS-amplicon sequencing, the latter approach was used to analyze temporal changes and specificity of OTUs to sponge species. A larger number of quality-filtered sequences were obtained from samples collected on the 4 May 2016 (average 13,136 sequences per sample; range 6,578–30,977) compared to the collection effort on the 13 November 2014 (4,062 sequences per sample; 64–16,199 range). Sequences were clustered into OTUs at a 97% similarity and non-fungal sequences were removed leaving a total of 155,298 sequences and 148 OTUs. Estimation of the Chao1, Good's coverage and Shannon's indices were conducted on normalized data (sub-sampled to 250 sequences), resulting in the removal of samples S_1_14, S_2_14, S_2_16, S_3_14, SW_1_14, and T_3_14 (sample-type_replicate-number_year) (Table 1). Good's coverage estimates were greater than 94% for all remaining samples, showing that the majority of OTUs were captured through the ITS-amplicon sequencing effort. Generally, Chao1 estimates positively correlated with the Shannon's diversity index.

Due to the filter feeding capacities of sponges, we expect the presence of "incidental" environmental fungi in our sponge samples. Fungal OTUs were therefore grouped into three categories; "occasional" (OTUs occurring only once in the six replicates per sample type), "variable" (OTUs occurring in two to five of the six replicates per sample type) and core OTUs (OTUs occurring in all six replicates per sample type). No core OTUs were observed in *C. concentrica*, *Scopalina* sp. and seawater, and only one core OTU was observed in *T. anhelans* (Table 2). Fungal communities of all three sponges were predominantly comprised of "variable" OTUs (>60% mean relative abundance).

Occasional OTUs were removed to create the "variable/core" fungal community dataset and was sub-sampled to a size of 250 reads, resulting in a total of 38 OTUs (Fig. 2). Similar to the whole community analysis (Table S2), Shannon's diversity indices of the "variable/core" communities were significantly lower in 2014 compared to 2016 (Table 3). "Variable/core" communities of *C. concentrica* and seawater were significantly different, but fungal diversity of *T. anhelans* was comparable between the two time points.

**Table 1 Number of reads, observed fungal OTUs, expected fungal OTUs (Chao1), Coverage (Good's coverage) and Shannon's index in seawater and sponge samples at a 97% sequences similarity threshold.**

| Sample name | Abbreviation | Total no. of quality-filtered reads | No. of fungal reads | Observed fungal OTUs | Expected OTUs (Chao1) ± SE (sub-sampled) | Good's coverage (sub-sampled) | Shannon's index (sub-sampled) |
|---|---|---|---|---|---|---|---|
| *C. concentrica* 1 2014 | C_1_14 | 16,199 | 14,849 | 8 | 3 ± 0 | 1.00 | 0.09 |
| *C. concentrica* 2 2014 | C_2_14 | 3,026 | 1,120 | 4 | 3 ± 0 | 1.00 | 0.90 |
| *C. concentrica* 3 2014 | C_3_14 | 6,198 | 4,808 | 4 | 1 ± 0 | 1.00 | 0.00 |
| *C. concentrica* 1 2016 | C_1_16 | 6,578 | 269 | 24 | 28 ± 18 | 0.97 | 1.94 |
| *C. concentrica* 2 2016 | C_2_16 | 21,700 | 3,437 | 27 | 10 ± 3 | 0.99 | 0.63 |
| *C. concentrica* 3 2016 | C_3_16 | 9,483 | 320 | 30 | 44 ± 35 | 0.94 | 2.07 |
| *Scopalina* sp. 1 2014 | S_1_14 | 1,532 | 49 | 2 | NA | NA | NA |
| *Scopalina* sp. 2 2014 | S_2_14 | 7,071 | 12 | 3 | NA | NA | NA |
| *Scopalina* sp. 3 2014 | S_3_14 | 1,992 | 107 | 4 | NA | NA | NA |
| *Scopalina* sp. 1 2016 | S_1_16 | 4,859 | 444 | 29 | 27 ±15 | 0.97 | 1.33 |
| *Scopalina* sp. 2 2016 | S_2_16 | 3,281 | 206 | 32 | NA | NA | NA |
| *Scopalina* sp. 3 2016 | S_3_16 | 24,565 | 665 | 36 | 39 ± 50 | 0.96 | 1.38 |
| *T. anhelans* 1 2014 | T_1_14 | 681 | 681 | 2 | 2 ± 1 | 1.00 | 0.03 |
| *T. anhelans* 2 2014 | T_2_14 | 6,947 | 6,703 | 9 | 5 ± 4 | 1.00 | 0.96 |
| *T. anhelans* 3 2014 | T_3_14 | 64 | 8 | 3 | NA | NA | NA |
| *T. anhelans* 1 2016 | T_1_16 | 19,481 | 18,300 | 21 | 5 ± 4 | 1.00 | 1.21 |
| *T. anhelans* 2 2016 | T_2_16 | 16,379 | 6,947 | 27 | 9 ± 17 | 0.99 | 0.77 |
| *T. anhelans* 3 2016 | T_3_16 | 30,977 | 30,956 | 19 | 3 ± 2 | 1.00 | 0.72 |
| Seawater 1 2014 | SW_1_14 | 1,066 | 2 | 2 | NA | NA | NA |
| Seawater 2 2014 | SW_2_14 | 3,379 | 3,314 | 2 | 2 ± 1 | 1.00 | 0.03 |
| Seawater 3 2014 | SW_3_14 | 2,647 | 1,588 | 3 | 1 ± 0 | 1.00 | 0.00 |
| Seawater 1 2016 | SW_1_16 | 6,139 | 3,379 | 33 | 12 ± 8 | 1.00 | 2.05 |
| Seawater 2 2016 | SW_2_16 | 12,439 | 5,576 | 45 | 21 ± 9 | 0.98 | 2.37 |
| Seawater 3 2016 | SW_3_16 | 25,069 | 19,496 | 40 | 14 ± 18 | 0.99 | 1.24 |

**Table 2 Number of total, transient, resident and core OTUs observed in sponge and seawater replicates.**

| Sample type | Total OTUs | Occasional OTUs | Variable OTUs | Core OTUs |
|---|---|---|---|---|
| *C. concentrica* | 52 | 24 (11.0% ± 5.2) | 28 (88.9% ± 13) | 0 (0%) |
| *Scopalina* sp. | 56 | 27 (35.7% ± 9.3) | 29 (64.2% ± 10.7) | 0 (0%) |
| *T. anhelans* | 49 | 32 (11.9% ± 5.4) | 16 (73.2% ± 10.2) | 1 (19.5% ± 35) |
| Seawater | 76 | 44 (5% ± 3.7) | 32 (94% ± 13.7) | 0 |

**Note:**
Numbers in brackets are the sum of the mean relative abundances of reads (percentages ± standard deviations).

Temporal samples were combined to compare the community diversity of sponge species and seawater to each other. "Variable/core" community of *Scopalina* sp. had the highest Shannon's index followed by *T. anhelans*, *C. concentrica* and seawater (Table 3), in contrast to the whole community analysis where the Shannon's index was highest for *Scopalina* sp., followed by seawater, *C. concentrica* and *T. anhelans* (Table S2). However,

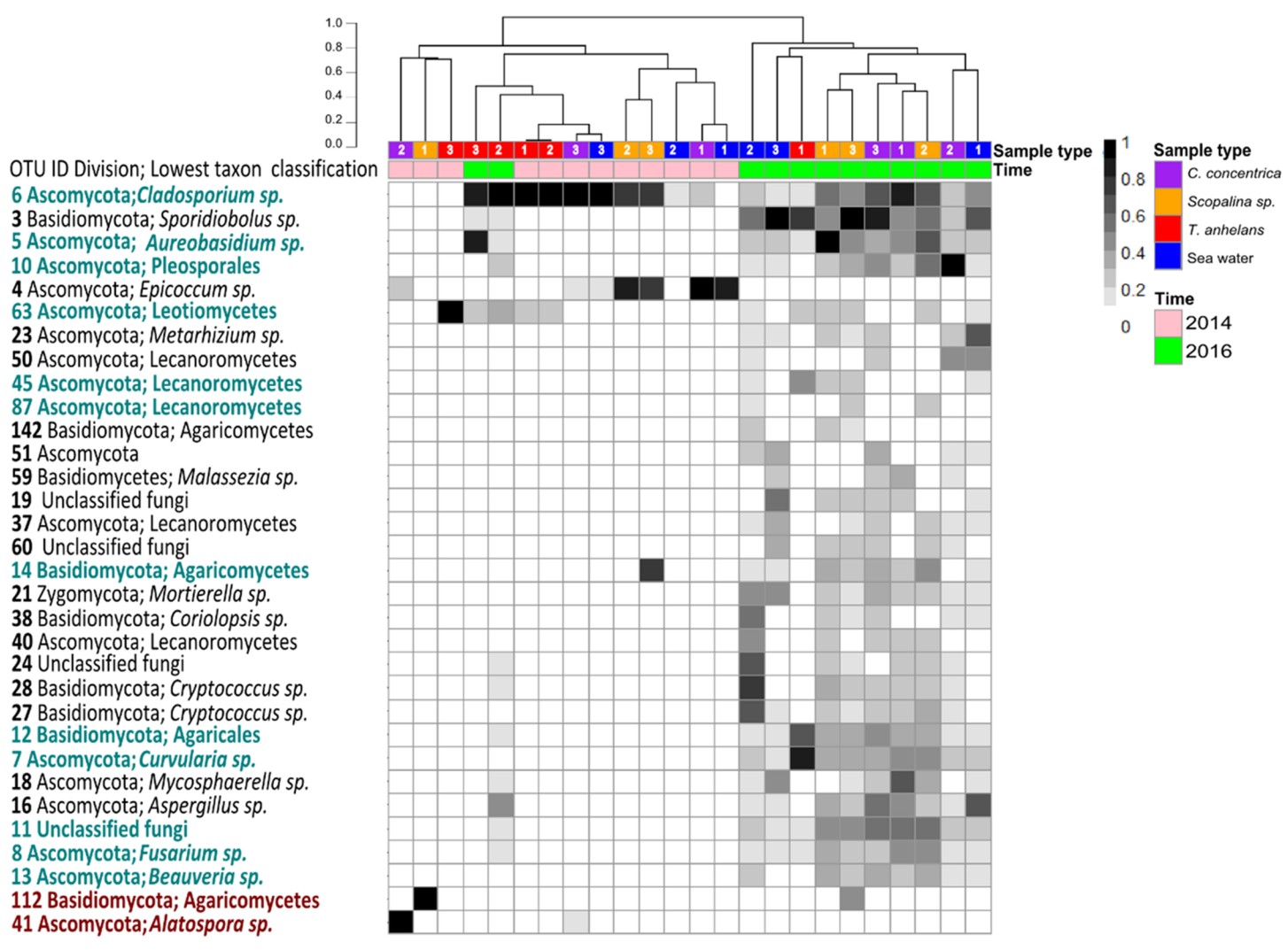

**Figure 2 Heatmap of "resident/core" sponge fungal communities, with OTUs only found in seawater removed.** Values are fourth roots of the relative abundance. Columns are clustered based on Bray–Curtis dissimilarity using hierarchical clustering with the "average" method (scale depicts percentage of dissimilarities). OTUs in teal indicate sponge enriched OTUs (relative abundances) compared to seawater and OTUs in maroon indicate OTUs present in sponges which were absent in seawater. Samples are indicated by the color purple: *C. concentrica*, orange: *Scopalina* sp., blue: seawater, and red: *T. anhelans*. Temporal samples are indicated by the color pink: 2014 and green: 2016. Numbers 1, 2 and 3 refer to individual replicates of each sample type.                               

Shannon's indices for both whole and "variable/core" communities for all samples were comparable to each other.

Fungal communities were overall quite variable in structure with average distances from the centroid calculated with PERMDISP being 0.57, 0.46, 0.44, and 0.58 for *C. concentrica*, *Scopalina* sp., *T. anhelans* and seawater, respectively. In contrast, the average distance to centroid for bacterial communities of *C. concentrica*, *Scopalina* sp., *T. anhelans* and seawater were relatively low with values of 0.21, 0.17, 0.11, and 0.22, respectively, and we saw a clear distinction in the composition and structure of these samples types (Fig. 3) Despite this relatively high variability in the fungal community,

**Table 3 Alpha diversity of "resident/core" fungal community as measured by the mean Shannon's index ± standard deviation (SD).**

| Temporal samples | Shannon's index (*P*-values) |
|---|---|
| 2014 vs. 2016 | 0.011 ± 0.014 vs. 1.35 ± 0.66 (**6.96e-05**) |
| *C. concentrica* 2014 vs. 2016 | 0.008 ± 0.015 vs. 1.41 ± 0.9 (**0.04**) |
| *Scopalina* sp. 2014 vs. 2016 | NA vs. 1.38 ± 0.0001 (NA) |
| *T. anhelans* 2014 vs. 2016 | 0.013 ± 0.018 vs. 0.75 ± 0.45 (0.62) |
| Seawater 2014 vs. 2016 | 0.013 ± 0.018 vs. 1.89 ± 0.39 (**0.01**) |
| **Sample type** | |
| Seawater vs. *Scopalina* sp. | 0.45 ± 0.52 vs. 1.38 ± 0.0001 (0.98) |
| *T. anhelans* vs. *Scopalina* sp. | 1.14 ± 1.06 vs. 1.38 ± 0.0001 (0.58) |
| *T. anhelans* vs. Seawater | 1.14 ± 1.06 vs. 0.45 ± 0.52 (0.59) |
| *Scopalina* sp. vs. *C. concentrica* | 1.38 ± 0.0001 vs. 0.71 ± 0.96 (0.77) |
| Seawater vs. *C. concentrica* | 0.45 ± 0.52 vs. 0.71 ± 0.96 (0.84) |
| *T. anhelans* vs. *C. concentrica* | 1.14 ± 1.06 vs. 0.71 ± 0.96 (0.96) |

Note:
Comparison of diversity between samples were calculated with ANOVA and multiple comparison with Tukey's test of subsampled resident/core fungal communities at an OTU-level clustered at 97%. *P*-values smaller than 0.05 are shown in bold.

beta-diversity analysis of the whole (Table S3) and "variable/core" (Table 4) fungal communities showed significant overall differences between samples in 2014 and 2016. *C. concentrica* and seawater community compositions (presence/absence) were also different between the two time points, in contrast to *T. anhelans*. However, temporal communities of sponges and seawater did not differ between the two time points based on relative abundances. Beta-diversity analysis further showed that the fungal communities of all three sponge species were comparable to the surrounding seawater. The only difference was observed between fungal community of *T. anhelans* and *Scopalina* sp., where whole community analysis showed differences only in their compositions, in contrast to the "variable/core" community analysis, where significant differences were seen both in terms of their structure and composition.

A total of 30 out of 32 OTUs observed in sponges were also present in the surrounding seawater, however 12 of these OTUs were enriched in sponge-associated communities (based on mean relative abundances) (Fig. 2 and Table S4). The majority of enriched OTUs were found in more than one sponge species: OTU_6, OTU_5, OTU_10, OTU_7, OTU_8, OTU_11, OTU_12, and OTU_13 were found in all three sponges, OTU_63 and OTU_45 were observed in *T. anhelans* and *Scopalina* sp., OTU_14 was observed in *C. concentrica* and *Scopalina* sp., and OTU_87 was observed in *Scopalina* sp. (Fig. 2). Two OTUs not found in seawater were OTU_41 (*Alatospora* sp.) (only observed in *C. concentrica*) and OTU_112 (belonging to the class Agaricomycetes and was unique to *Scopalina* sp.), and both were highly variable in abundance and sometimes absent in one or more replicate samples. Fungal communities in sponges were generally dominated by a few OTUs. The top seven, eight, and four most abundant OTUs made up ~90% of the total relative abundance of *C. concentrica*, *Scopalina* sp. and *T. anhelans*, respectively. The most abundant OTU observed across all samples was OTU_6 (*Cladosporium* sp.),

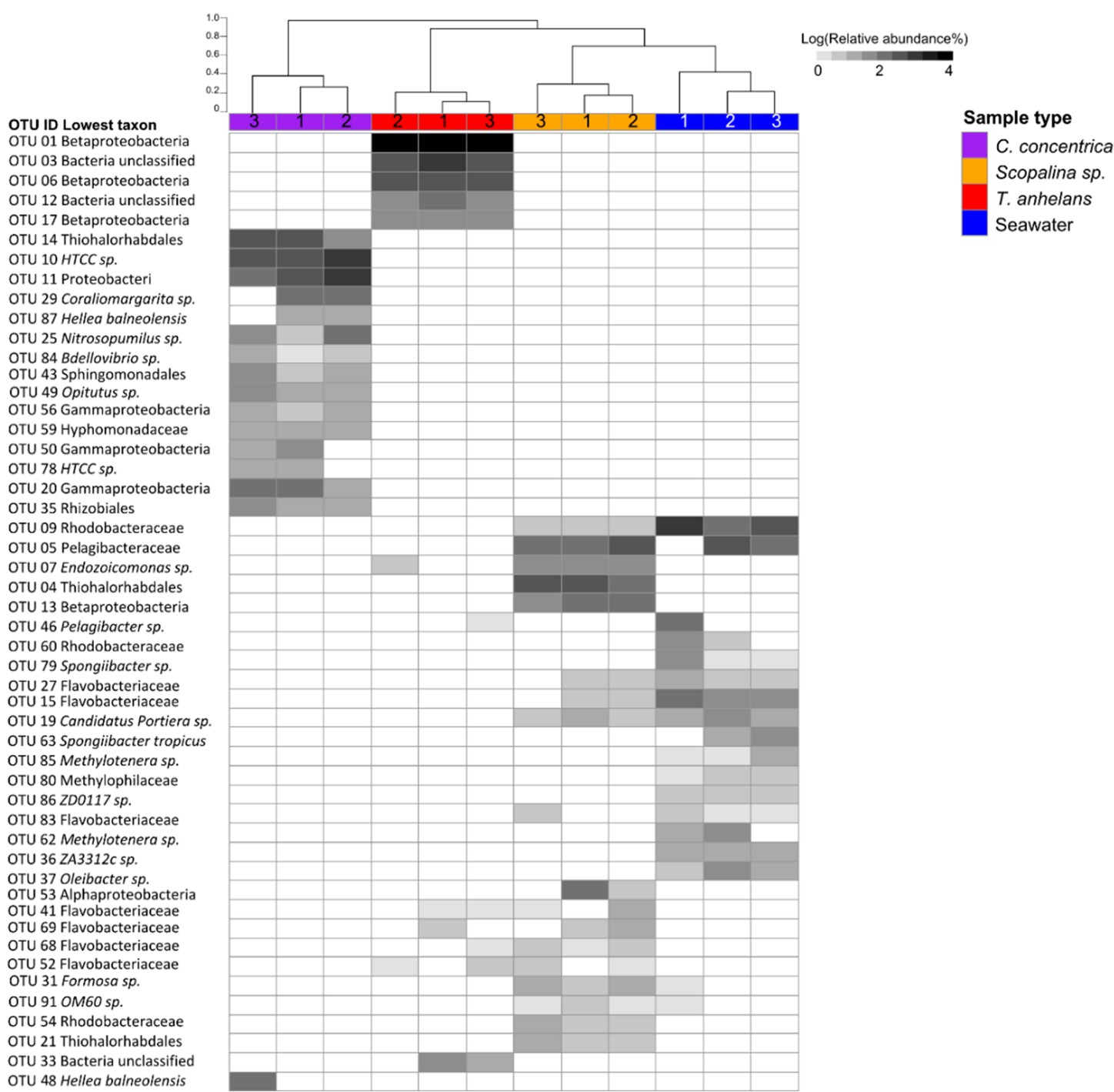

**Figure 3 Heatmap of the bacterial community composition based on 16S rRNA gene sequences for the sponge samples collected in 2014.** OTUs were clustered at 97% similarity and the lowest taxonomy classification are given. Columns are clustered based on Bray–Curtis dissimilarity using hierarchical clustering with the "average" method (scale depicts percentage of dissimilarities). Samples are indicated in purple: *C. concentrica*, orange: *Scopalina* sp., blue: seawater, and red: *T. anhelans*. Numbers 1, 2, and 3 indicate sample replicates.

**Table 4 PERMANOVA analysis (based on relative abundance and presence-absence values) of "resident/core" fungal communities (normalized) at an OTU-level cluster at 97% similarity of sponges and seawater samples.**

| Temporal samples | *P*-value base on relative abundance | *P*-value based on presence-absence |
|---|---|---|
| 2014 vs. 2016 | **0.023** | **0.001** |
| *C. concentrica* 2014 vs. 2016 | 0.1 | **0.001** |
| *Scopalina* sp. 2014 vs. 2016 | NA | NA |
| *T. anhelans* 2014 vs. 2016 | 0.40 | 0.40 |
| Seawater 2014 vs. 2016 | 0.1 | **0.008** |
| **Sample type** | | |
| *C. concentrica* vs. *Scopalina* sp. | 0.4 | 0.386 |
| *C. concentrica* vs. *T. anhelans* | 0.24 | 0.154 |
| *C. concentrica* vs. seawater | 0.54 | 0.632 |
| *Scopalina* sp. vs. *T. anhelans* | **0.048** | **0.048** |
| *Scopalina* sp. vs. seawater | 0.5 | 0.338 |
| *T. anhelans* vs. seawater | 0.143 | 0.139 |

**Note:**
*P*-values smaller than 0.05 are shown in bold.

but was most abundantly found in *T. anhelans* with a relative abundance range of 0–99% across six individual sponge samples. Only one core OTU was found in the sponge *T. anhelans* (OTU_63) identified as belonging to the class Leotiomycetes.

# DISCUSSION

## Assessment of fungal diversity in sponges

Sponge-associated fungal communities have been studied through culture-dependent (*Henríquez et al., 2014*; *Liu et al., 2010*; *Wang, Li & Zhu, 2008*; *Höller et al., 2000*; *Proksch et al., 2008*; *Wiese et al., 2011*; *Yu et al., 2013*; *Li & Wang, 2009*) and culture-independent methods (*Gao et al., 2008*; *Naim, Smidt & Sipkema, 2017*; *He et al., 2014*; *Passarini et al., 2015*; *Rodríguez-Marconi et al., 2015*; *Wang et al., 2014*). However, no study has so far combined both approaches to provide insights into the extent to which the two community assessments capture sponge-associated diversity.

In this study, 42 fungal isolates were classified into eight different OTUs (*Penicillium* spp. (3), *Togniniaceae*, *Acostalagmus* sp., *Trichoderma* sp., *Cladosporium* sp., and *Aspergillus* sp.). Three isolates of the genus *Penicillium* (Eurotiales) were also found in the ITS amplicon sequencing and one isolate (OTU_119) was cultivated from all sample types. *Penicillium* can be considered as a "generalist" (or core organism) and is an ubiquitous fungal genus commonly cultured from various environments, including sponges from around the world (*Gao et al., 2008*; *Höller et al., 2000*; *Li & Wang, 2009*; *Liu et al., 2010*; *Passarini et al., 2013, 2015*; *Paz et al., 2010*; *Pivkin et al., 2006*; *Proksch et al., 2008*; *Wang, Li & Zhu, 2008*). Fungi from the family *Togniniaceae* (order Diaporthales) are likely of terrestrial origins as a common genus (*Togninia*) within is often associated with wilt and disease of woody plants (*Crous et al., 1996*; *Rossman, Farr & Castlebury, 2007*).

*Acrostalagmus* (order Hypocreales) contains pathogenic fungi against animals, plants and humans (*Kubicek, Komon-Zelazowska & Druzhinina, 2008*). Some species of this genus have been isolated from deep-sea sediments and have been reported to produce anti-tumor (*Wang et al., 2012*) and antifungal compounds (*Sato & Kakisawa, 1976*; *Ellestad, Evans & Kunstmann, 1969*). *Togniniaceae* and *Acrostalagmus* sp. so far have not been reported to be cultured from marine sponges, while *Trichoderma* sp., *Aspergillus* sp. and *Cladosporium* sp. have previously been cultivated from various marine sponges (*Proksch et al., 2008*; *Höller et al., 2000*; *Wang, Li & Zhu, 2008*; *Paz et al., 2010*; *Passarini et al., 2013*).

ITS-amplicon based community profiling captured more fungal diversity (26 OTUs) compared to cultivation (eight OTUs), which is consistent with previous studies examining fungal communities in marine sediments, decaying leaves and soil (*Jebaraj et al., 2010*; *Nikolcheva, Cockshutt & Bärlocher, 2003*; *Borneman & Hartin, 2000*; *Singh et al., 2012*). However, similar to the studies of *Jebaraj et al. (2010)* and *Singh et al. (2012)*, the cultivation yielded also distinct isolates (five) that were not found through the culture-independent method. Three of the unique isolates were cultivated from *T. anhelans* (OTU_266, OTU_268, and OTU_269), one from *Scopalina* sp. (OTU_265) and OTU_264, which was cultivated from *C. concentrica*, *Scopalina* sp. and seawater. This occurrence could be explained by the overall low number of fungal reads obtained for *T. anhelans* and *Scopalina* samples in 2014, when the cultivation was also performed (Table 1). Low numbers of fungal reads due to the amplification of non-target (e.g., sponge) DNA was also noted as a challenge in previous fungal diversity studies (*He et al., 2014*; *Naim, Smidt & Sipkema, 2017*; *Gao et al., 2008*). Another explanation for the discrepancy between the two methods is that the isolates could belong to the rare biosphere, which is supported by the fact that three of the isolates were unique (i.e., only cultured once) (Table S1). Similarly, the three *Penicillium* spp., which were found by both methods, were only found in low relative abundances in the ITS-amplicon sequencing data. This indicates that fungal cultivation, even under the various conditions tried here, does not capture abundant community members in sponges, but rather those that are relatively rare. This situation is analogous to what has been seen in many studies examining bacterial communities in sponges (*Webster & Hill, 2001*; *Esteves et al., 2016*; *Hentschel et al., 2001*; *Muscholl-Silberhorn, Thiel & Imhoff, 2008*; *Li, He & Miao, 2007*). Our results indicate that ITS-amplicon-based community profiling therefore likely provides a more realistic assessment of fungal diversity in sponges and thus should be seen as the "gold standard" for community assessment, similar to what has been proposed for fungal diversity research in other environments (*Pang & Mitchell, 2005*; *Jeewon & Hyde, 2007*).

## Host specificity of sponge-associated fungi

*Li & Wang (2009)* have previously classified fungi found in sponges into three groups: "sponge-generalists" (i.e., found in all sponge species), "sponge-associates" (i.e., found in more than one sponge species), and "sponge-specialists" (i.e., found in only one sponge species). This classification is very limited and dependent on (a) the number of sponge

species studied, (b) the methods used (culture-dependent or -independent approach (see above), and (c) experimental design (use (or lack) of biological replicates or seawater reference communities). These limitations have been highlighted by *Yu et al. (2013)* in their summary of sponge-associated fungi reported since 1996. The summary illustrates differences in classification of the same fungal genus in different studies. For example, the fungal genera *Fusarium* and *Trichoderma* were classified as "sponge generalists" according to *Menezes et al. (2010)*, but were classified as "sponge-associates" by *Höller et al. (2000)* and *Li & Wang (2009)*, and were completely absent in various sponges investigated by *Gao et al. (2008)* and *Pivkin et al. (2006)*. Therefore, one should be careful when using these terminologies for host-specificity of fungal genera. Another challenge in determining the specificity of sponge-associated fungi is the filter-feeding capacity of sponges that brings large amounts of seawater along with all its constituents into the sponge tissue. Studies that are based on the once-off, presence-absence measurement of isolates (*Thirunavukkarasu et al., 2012*; *Paz et al., 2010*; *Liu et al., 2010*; *Li & Wang, 2009*) or sequences (*Gao et al., 2008*; *He et al., 2014*; *Zhang et al., 2016*; *Rodríguez-Marconi et al., 2015*) to support claims of specificity, could therefore be misleading as fungi found inside the sponge might simply just be accidentally trapped, rather than having any meaningful interaction with the sponge. Indeed, studies on bacterial communities in sponges have found that ecologically important organisms are consistently present in the sponge and usually at high relative abundances (*Thomas et al., 2016*; *Fan et al., 2012*; *Hentschel et al., 2002*, *2003*; *Lee et al., 2011*; *Wilkinson & Fay, 1979*). Therefore, we propose and recommend implementing a different guideline to the above classifications, where "occasional" and "variable/core" fungi can be defined by examining biological replicates over time. "Variable/core" fungi that are found in combination with high relative abundances compared to the surrounding environment (i.e., seawater) should then be considered "sponge-enriched". Implementing these guidelines, our study revealed that "variable/core" fungal communities in sponges have low diversity ($\leq 28$ fungal OTUs were observed in each sponge species), high variability and that the majority of fungal OTUs were not specifically enriched in any of the sponge species investigated here. These findings are consistent with a recent study by *Naim, Smidt & Sipkema (2017)*. In contrast, the bacterial communities of the sponges studied here have previously been shown to have diverse and host-specific sponge-enriched bacterial phyla, which was consistent over time and space (*Fan et al., 2012*; *Esteves et al., 2016*). Furthermore, bacterial communities had low variability and were distinct from the bacterial community in the surrounding seawater (*Thomas et al., 2016*; *Fan et al., 2012*). Indeed we could reproduce this low variability and distinctive composition of bacterial communities (see Fig. 3) using the same technical approaches and DNA extracts used for the fungal analysis, indicating that the high variability in the fungal communities truly reflects fundamental differences in the ecology of these two microbial groups.

Fungal communities in the three sponges studied here appear to be largely influenced by the community of the surrounding seawater. This is broadly reflected in the higher fungal diversity observed in 2016 compared to 2014, in both the seawater and in all sponges. However, the degree of influence by the surrounding seawater appears to differ

for different sponge species. Fungal communities in *T. anhelans* did not significantly differ between the two time points, in contrast to *C. concentrica* and seawater. Additionally, fungal communities of *T. anhelans* and *Scopalina* sp. were found to be significantly different between each other suggesting that the two sponge species may exert different selective pressures on their fungal communities. Differences in the relative abundances of the same OTUs in the sponges further support the occurrence of preferential interactions of fungi with different sponge species. For example, OTU_10 was found in *T. anhelans*, *Scopalina* sp. and *C. concentrica* with mean relative abundance of 0.09% ± 0.21 (range of 0–0.43%), 1.8% ± 3.6 (range of 0–9.2%), and 16.7% ± 38 (range of 0–94.7%), respectively.

## Potential roles of sponge-associated fungi

Seven out of the 12 "sponge-enriched" OTUs could not be classified to a genus level, which indicates that they are either novel, yet-to-be-studied organisms or their ITS sequences are not available in the databases we used (Table S5). The seven OTUs could be identified to the lowest taxonomic classification as: kingdom fungi (one), class Lecanoromycetes (two), Leotiomycetes (one), Agaricomycetes (one) and within the class Agaricomycetes the orders Pleosporales (one) and Agaricales (one). The other five sponge-enriched OTUs were classified as *Cladosporium* sp. (OTU_6), *Fusarium* sp. (OTU_8), *Aureobasidium* sp. (OTU_5), *Curvularia* sp. (OTU_7), and *Beauveria* sp. (OTU_13). OTU_112 and OTU_41 were found exclusively in *Scopalina* sp. and *C. concentrica*, respectively. OTU_112 was classified to belong to class Agaricomycetes, which contains saprotrophs, pathogens and mutualists (*Hibbett et al., 2014*). Agaricomycetes were found frequently in this study, in fact eight out of 10 Basidiomycota fungi observed belong to this class. Sponge-enriched order Agaricales and Polyporales (within class Agaricomycetes) have also been detected in other marine sponges (*He et al., 2014*; *Naim, Smidt & Sipkema, 2017*) and corals (*Amend, Barshis & Oliver, 2012*). OTU_41 (*Alatospora* sp., order Leotiales) has not been previously reported in marine sponges. *Alatospora* spp. are commonly associated with decaying wood/leaves in fresh water streams (*Das, Royer & Leff, 2008*; *Hosoya & Tanaka, 2007*; *Sridhar & Kaveriappa, 1989*). Together this shows that fungi with potential saprophytic properties are widespread in sponges and indicates that they may exploit the nutrient rich environment of the sponge host and/or could contribute to nutrient uptake via the breakdown of plant-derived detritus or plankton filtered from the surrounding seawater (*Zhang et al., 2006*; *Hyde et al., 2013*; *Hibbett et al., 2014*).

One core OTU (OTU_63) detected in *T. anhelans* was classified belonging to the class Leotiomycetes. Interestingly, Leotiomycetes were prevalently cultivated (75% of total isolates) from Antarctic sponges, where a third of the fungal isolates were classified to the genus *Geomyces* and had strong antimicrobial activity. Another third of the isolates could not be identified to a genus level (*Henríquez et al., 2014*), indicating that this class has much unexplored genus-level diversity. More studies on Leotiomycetes in the future are essential to help elucidate its potential ecological function in sponges.

The class Lecanoromycetes (mostly enriched in *T. anhelans*) contains most of the lichen-forming fungal species (*Miadlikowska et al., 2006*). An OTU belonging to the order Telochistales (class Lecanoromycetes) was also found in the sponge *Halichondrida panicea* (an encrusting sponge) and was closely related to a lichen-forming fungal isolate (*Naim, Smidt & Sipkema, 2017*). The lichen-forming genus *Koralionastes* has been further reported to have a unique physical association with crustaceous sponge and was postulated to be nutritionally dependent on the sponges (*Kohlmeyer & Volkmann-Kohlmeyer, 1992*). Lichen have been reported to facilitate the destruction of rocks and provide the adjacent water layers with nutrients and trace elements that can benefit hydrobionts, including sponges (*Kulikova et al., 2013*). This raised the possibility that the presence of Lecanoromycetes in *T. anhelans* (relative abundance range of 0–0.33%) and in sponge *H. panicea* may arise from a close co-existence of crustose lichen and encrusting sponges on rock fragments (*Suturin et al., 2003*).

OTU_6 (*Cladosporium* sp.) and OTU_8 (*Fusarium* sp.) belong to genera that are known to be saprotrophs, have previously been isolated from various marine sponges (*Menezes et al., 2010*; *Wang, Li & Zhu, 2008*; *Paz et al., 2010*; *Thirunavukkarasu et al., 2012*; *Zhou et al., 2011*; *Liu et al., 2010*; *Ding et al., 2011*; *Li & Wang, 2009*; *Proksch et al., 2008*) and have been reported to produce bioactive compounds (*Gesner et al., 2005*; *Wiese et al., 2011*; *Jadulco et al., 2002*). For example, *Cladosporium herbarum* isolated from the sponges *Callyspongia aerizusa*, *Aplysina aerophoba* and *Callyspongia aerizusa* produced various different antimicrobial compounds, such as acetyl Sumiki's acid (*Gesner et al., 2005*), pyrone derivatives and macrocyclic lactones (*Jadulco et al., 2002*). The marine-derived *Cladosporium* sp. F14 has also been found to produce compounds, which inhibited larval settlement of the bryozoan *Bugular neritina* and the barnacle *Balanus amphitrite* (*Qi et al., 2009*). *Fusarium* spp. isolated from the sponge *Tethya aurantium* were found to produce antibacterial and insecticidal compounds, such as equistins and enniatine (*Wiese et al., 2011*). *Cladosporium* sp. and *Fusarium* sp. may similarly produce secondary metabolites in the sponges investigated here, which may potentially contribute to sponge host defense.

OTU_5 (*Aureobasidium* sp., order Dothideales), OTU_7 (*Curvularia* sp., order Pleosporales) and OTU_13 (*Beauveria* sp., order Hypocreales) had closest sequence similarity to common terrestrial species (Table S5), which can be facultative pathogens of plants, humans and insects (*Zalar et al., 2008*; *Rinaldi et al., 1987*; *Liu et al., 2009*; *Rehner et al., 2011*). All three genera have been previously isolated from various sponges around the globe (*Li & Wang, 2009*; *Wang, Li & Zhu, 2008*; *Gao et al., 2008*; *Passarini et al., 2015*; *Henríquez et al., 2014*; *Höller et al., 2000*; *Wiese et al., 2011*). Sponges have been reported to harbour various other fungal species related to terrestrial plant, human and animal pathogens, such as *Aspergillus terreus* and *Cladosporium tenuissimum* (*Zhou et al., 2011*; *Liu et al., 2010*). Additionally, the sponge species *Spongia obscura* and *Ircinia strobilina* have been reported to host the fungus *Aspergillus sydowii*, which is a causative agent of a disease in the Caribbean sea fan corals (*Gorgonia* spp.) but had no notable affect to the sponge health (*Ein-Gil et al., 2009*). This suggests that sponges could be reservoirs of potential marine and terrestrial pathogens and may provide an environment for fungal propagule survival and dispersal.

## CONCLUSION

Our study shows that the sponge samples analyzed here contain phylogenetically diverse fungi (eight fungal classes were observed) that have low host-specificity and broadly reflected communities within the seawater. The same or highly similar fungal species have been detected in sponges around the world, which suggests a prevalence of horizontal transmission where selection and enrichment of some fungi occur for those that can survive and/or exploit the sponge environment. Our current sparse knowledge about sponge-associated fungi indicates that fungal communities may perhaps not play as an important ecological role (beside the exceptional case reported of vertically transmitted endosymbiotic yeast; *Maldonado et al., 2005*) in the sponge holobiont compared to bacteria or archaea. However, the interaction between sponges and fungi provides another layer to the already complex sponge environment, which may drive the evolution of not only bacteria and archaea (*Thomas et al., 2016*; *Hentschel et al., 2002*; *Fan et al., 2012*; *Thomas et al., 2010*), but possibly also fungi. The ecology and function of sponge-associated fungi represents a frontier of microbial diversity research awaiting further studies.

## ACKNOWLEDGEMENTS

The authors would like to thank Dr. Shaun Nielsen and Dr. Cristina Diez-Vives for help with sponge sampling.

### Funding

This work was supported through the Australian Research Council and the Betty and Gordon Moore Foundation. Mary Nguyen was supported by a Postgraduate scholarship from the Australian Government. The funders had no role in study design, data collection and analysis, decision to publish, or preparation of the manuscript.

### Grant Disclosures

The following grant information was disclosed by the authors:
Australian Research Council and the Betty and Gordon Moore Foundation.
Australian Government.

### Competing Interests

Torsten Thomas is an Academic Editor for PeerJ.

### Author Contributions

- Mary T.H.D. Nguyen conceived and designed the experiments, performed the experiments, analyzed the data, prepared figures and/or tables, authored or reviewed drafts of the paper, approved the final draft.
- Torsten Thomas conceived and designed the experiments, analyzed the data, authored or reviewed drafts of the paper, approved the final draft.

## Field Study Permissions

The following information was supplied relating to field study approvals (i.e., approving body and any reference numbers):

Sampling of sponges was performed under the scientific collection pemit P13/0007-1.1 issued by the New South Wales Department of Primary Industries.

## DNA Deposition

The following information was supplied regarding the deposition of DNA sequences:

Raw ITS-amplicon sequences are available through the NCBI Sequence Read Archive (SRA) under Bioproject ID: PRJNA419577, accession number SRP125576.

## Data Availability

The scripts used in this study, and the commands for the entire ITS amplicon and 16S rRNA gene analysis, are available in the Supplemental Information.

## Supplemental Information

Supplemental information for this article can be found online at http://dx.doi.org/10.7717/peerj.4965#supplemental-information.

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
