# Peer review of "Diversity, host-specificity and stability of sponge-associated fungal communities of co-occurring sponges"

_PeerJ, doi:10.7717/peerj.4965_

## Round 0.1 · original submission · Major Revisions

I would modulate the use of "over time " throughout the manuscript, such as in the statement in the abstract in which you refer than in your study you assessed the fungal community composition of three co-occurring sponges and the surrounding seawater over time. You sampled 2 years and over time would refer more to a time series experiment that is not the case of your study.

It seems that the sequencing data from the first sampling performed in 2014 although was obtained using same technology than for 2016 it had very low number of reads and I would consider if some samples should be included due to extremely low number of reads.

Give reference for primers ITS1f-F (5’ TTGGTCATTTAGAGGAAGTAA 3’) and reverse ITS4 (5’ 160 TCCTCCGCTTATTGATATGC 3’) White et al....

Why did you use only R1 reads. Using R2 reads that will cover part of the ITS2 region can provide also nice information, especially since ITS2 is shown to be more variable than ITS1 region.

Concerning Sequence analysis of 16S rRNA community profiling, where are these results? I see only one sentence L289-L291! You rather remove it or please, try to get some results and modify the discussion appropriately.

Please review the misspelling use of OUT instead of OTU, especially for supplementary tables.

I cannot open any of the TXT files of supplementary material.

I will modulate the generalization that the results are globally representative for sponge-associated fungi since you just sample a few specimens in a very particular location.

·

Basic reporting

The manuscript by Nguyen and Thomas explores the temporal stability of fungi associated to three different marine sponges and compares data with surrounding seawater communities and fungi recovered from the same samples through isolation. There are few similar papers with replicates on sponge-fungal communities. Therefore the manuscript adds substantially to the state of the art of the topic.
The study design is appropriate, as are the methods used, the paper is well-written and therefore I think it deserves publication in peerJ.

I do have minor comments specified below and a number of comments to be more cautious with the presentation and discussion of results (in general comments to the authors):

Experimental design

see comment in box 1.

Validity of the findings

see comments in box 1 and some specific remarks related to this in "general comments for the authors"

Additional comments

MATERIALS AND METHODS
l120: "pemit" = "permit"
l124: what are "fungal samples". At this stage, it is not clear where they came from or what they are. Do you mean the sponge samples harbouring the fungi?
l125: 200 ml of seawater was filtered. This is quite a bit less than most studies do for seawater? Why so little?
l155: was it really 8 minutes?
l159: is there a reference to the ITS primers used?
l190: based on the title, introduction and aim, this section comes as a surprise. It would need some reasoning in the intro why to include this or to leave it out.
Or is it a cut and paste error...?

RESULTS
l267: I understand the categorisation made, but I am not sure if it works out well. Since all samples were collected from a small space, it is very likely that the same 'environmental' fungi will be found in multiple replicates and therefore they may just be transient in many sponge individuals. Here it would be important to compare the fungi found in the sponges to the fungi found in the seawater to help to decide what is transient, resident and 'core'.
And if you use the classification, it would be good to have it consistently recurring throughout the manuscript. For example, now in the conclusion there are hints towards these terms, but they are not actually mentioned anymore.
l306: "OUT_8"= "OTU_8"
l311-313: the way it is stated is not wrong, but it would be more informative to also report that many individuals of these species did not harbour this fungal OTU, because the high averages are solely based on very high relative abundance in one sample for each of the 2 OTUs.
l317: the results should be presented a bit more cautiously here. Here differences in relative abundances are compared, but one sample (T. anhelans 3, 2014) that causes the high relative abundance for OTU63 in this sponge species is based on 8 fungal reads, which makes the relative abundance kind of meaningless. Please also check other number that are presented here in the results, because this one creates a false idea of "difference".

DISCUSSION
l334: "Penicillium can be considered as a 'generalist'...." Would you also imply it makes part of the 'core' fungi isolated? I'm exploring how this matches with the terms you introduced for the paper.
l407: "only other study...". There is now also the study of Chaib De Mares et al (2017) where fungal communities in replicates in Aplysina aerophoba, Aplysina cauliformis, Dysidea avara and Dysidea etheria.
Front. Microbiol. 8, doi: 10.3389/fmicb.2017.02560
l430: "OUT_8" = "OTU_8"
l431: "OUT_13" = "OTU_13", also OUT41
l466-481: in the discussion data from fungi isolated from sponges are mixed in with speculation on ecological function. This should be done with caution. Also from this study it is shown that there is near to no overlap between the cultivated fungi and fungi observed through cultivation-independent techniques.
Also line 487.
l506: correct the formatting

·

Basic reporting

The authors describe fungal communities that are associated to three distinct marine sponges. They show that only a small fraction of the fungal community might be species-specific and that various species are accumulated from the surrounding seawater. Moreover, they show that cultivation-dependent and –independent approaches result in the identification of different fungal lineages. The manuscript is well-written and requires minor clarifications/amendments.

Experimental design

The experiments are well described and concise for most parts. Sequencing of fungal ITS regions was conducted in 2014 and 2016. Was the same instrument and chemistry used for both approaches? Different approaches would explain the variation in the results from these experiments. The Material & Methods section includes the description of an amplicon study with 16S gene fragments; however the results from this approach were later not included in the Results and Discussion sections.

Validity of the findings

The findings are sound and well described.

Additional comments

L27: Correct ‘that have so far have not been’ and ‘with in’ in the two sentences.
L130: Brackets are only required for the year of the reference.
L136 & 138 and other parts: Include a separator between values and units.
L73-174: Was the same instrument and chemistry used for both approaches?
L190-207: Why are the results not presented later on?
L229 and other parts: Most journals require italicized writing of all taxonomic ranks with a Latin name.
L289-291: This is the only passage that mentions results from the 16S RNA gene fragment assessment.
L306: ‘OTU’
L308: ‘OTU_14 was’
L314: Rephrase this sentence for better understanding.
L319 and other parts: For taxa with a high variation in abundance it would be better to present the range, e.g. x% - y%.
L359-360: This doesn’t explain why this was not a problem in the approach conducted in 2016.
L386: add brackets for the year of publication
L426: It could also be that these are species without database entries in the utilized reference database or generally without sequenced ITS region although the species might be described already.
L431: ‘OTU’
L461-464: Is it plausible that a lichen-forming fungus occurs at the sampling site? It might be a close relative or a non-lichenized lineage.

Reviewer 3 ·

Basic reporting

No comment

Experimental design

No comment

Validity of the findings

The manuscript generally complies with the editorial criteria: the research is within the scope ofthe Journal, poses meaningful questions and provides a broad discussion with the knowledge in the field.
It is well written, adequately referenced and appropriately structured with data properly considered and nicely presented. This is in dissonance with the fact that the experimental data obtained in two separated sets are not sufficient to validate author’s generalized interpretations.

Additional comments

In this paper, the authors describe the diversity of three co-occurring sponge’s associated fungi, evaluate their incidence stability and host specificity.
Since the knowledge about fungi in these environments is rather scarce the work of Nguyen and Thomas is of great interest for marine ecologist and for biodiversity-interested individuals in general.
The work presented adds valuable information towards the current knowledge in the field of marine ecology and fungal diversity. The comprehensive introduction and competent discussion make the article newsworthy to read. Nonetheless, the results are over-interpreted and conclusions insufficiently supported.
I have the following questions:
-The authors declare to present the first data set with biological replications. Yet, 4 out of 6 Scopalina samples were represented by very low number of reads and practically it is very difficult to treat them as decent biological replicas.
-What is the reason for such a big difference in NGS results standard between data from 2014 and 2016? I am not sure if such data should be concurrently analysed (even after normalization), as can be valued in respective graphics/tables where 6 of them remain NA. The paper would gain a lot if NGS procedure for 2014 samples could be repeated.
-What about the technical repetitions - what is now the standard in such research? Analysing data presented in Tab.1 it seems that ITS PCR amplifications were heavily biased, especially for 2014 samples (e.g., 3 OTUs out of 8 reads vs 2 out of 681 in T. anhelans case). Although in their Discussion authors consider the problem of PCR nonspecific starters for fugal symbionts, lack of amplifications technical details make respectable explication impossible. Were PCR amplifications performed in several independent reactions for each DNA sample? The extremely low fungal reads obtained suggest single reaction.
-Why the cultivation of sponge-associated fungi was not performed in 2016 as well? Since cultivation recovered some organisms not detected by culture-independent approach, these data would strengthen validation of the statements about the in-time stability of fungal population for a given sponge.

Some clarifications in Material and Methods section are needed:
- how many DNA extractions from each sample were done?,
- what was the weight of sponge samples?,
- was the NGS sequencing done in technical repetitions?

---

## Round 0.2 · accepted · Accept

All reviewers and myself considered that you have revised your manuscript satisfactorily.

# ·

Basic reporting

Thanks for the revised version.
Main comments were well addressed.
Some small things listed below.

best wishes

Experimental design

Original Comment:
l190: based on the title, introduction and aim, this section comes as a surprise. It would need some reasoning in the intro why to include this or to leave it out.
Or is it a cut and paste error...?

Response:
We provide now a brief motivation why we also analysed the bacterial community composition:” Given the high variability of fungal communities observed between biological replicates (see below), we wanted to understand if this is a peculiar aspect of the samples we took and how we processed them or if this is due to real biological variation. We therefore also analysed all samples for the bacterial community composition, which has been shown to be very consistent between replicates of the three sponges analysed here (Fan et al. 2012, Esteves et al. 2016).

Comment revision
Alright. That makes sense and I agree. I see that they are also shown now in the results section and included in the discussion.
Please also upload the 16S data then to NCBI or elsewhere so they are of use to the community. As you mentioned before, sponge sample for which both fungi and bacteria were sequenced are rare.

Validity of the findings

Original Comment:
l267: I understand the categorisation made, but I am not sure if it works out well. Since all samples were collected from a small space, it is very likely that the same 'environmental' fungi will be found in multiple replicates and therefore they may just be transient in many sponge individuals. Here it would be important to compare the fungi found in the sponges to the fungi found in the seawater to help to decide what is transient, resident and 'core'.
And if you use the classification, it would be good to have it consistently recurring throughout the manuscript. For example, now in the conclusion there are hints towards these terms, but they are not actually mentioned anymore.

Response:
The definition of “core” is typically applied to OTUs that are found in all (or a very large proportion) of replicate samples for a particular sample type or habitat (e.g. Schmitt et al. ISME J. 2012, 6(3): 564–576.). It is independent of the core organisms being also found in (or derived from) other habitats or sample types. We therefore think the term is appropriate in the current context. We used the term “transient” to specify OTUs that are only once found in any samples type, however we agree that this term has an unintended connotation of a particular kind of interaction. We therefore renamed this category as “occasional”. The same applies for “resident”, which might also suggest some ecological interaction, and our definition of being found more than once, but not in all samples, is better described with the term “variable”. This definition allowed us to focus our analysis on OTUs, for which we have evidence of repeated occurrence in each sample type.

We do also compare those ‘variable/core’ OTUs found in sponges to those found seawater (see Figure 2), and use the category definition in the discussion to conclude that “Implementing these guidelines, our study revealed that ‘variable/core’ fungal communities in sponges have low diversity (≤28 fungal OTUs were observed in each sponge species), high variability and that the majority of fungal OTUs were not specifically enriched in any of the sponge species investigated here.”

Comment revision
The authors have changed the terminology, which is ok to me, but not really necessary to me as it does not really change it. So for the final version I would like to leave it up to the authors which term they prefer.
In their reply, the authors state that they adopt the definition of Schmitt et al (ISME 2012) and that is fair. I’m not a fan of this definition, but that should not prevent publication. I’m not a fan of it, because I think it has many of the same drawbacks that the authors later use in their discussion on the terminology of ‘sponge generalists’ and ‘sponge specialists’: (l389-392 of track changes version) “This classification is very limited and dependent on a) the number of sponge species studied, b) the methods used (culture-dependent or -independent approach (see above), and c) experimental design (use (or lack) of biological replicates or seawater reference communities).”
But as mentioned before, I think my personal view here should not prevent publication and I know that many don’t share my view here.

·

Basic reporting

The manuscript fulfills all criteria.

Experimental design

The manuscript fulfills all criteria.

Validity of the findings

The manuscript fulfills all criteria.

Additional comments

The authors have answered all arising questions and they have implemented the recommended changes.

Reviewer 3 ·

Basic reporting

The manuscript by Nguyen and Thomas adds valuable information towards the current knowledge in the field of marine ecology and fungal diversity.
It is well written, adequately referenced and appropriately structured with data properly considered and nicely presented.

Experimental design

The experiments are now described with sufficient detail.

Validity of the findings

The presnted manuscript broaden the picture of diversity of fungi associated to marine sponges.
In the revised manuscript conclusions are properly linked and limited to the presented results.

Additional comments

The revised manuscript now properly points the area of presented study. I found very sound the addition of data for general 16SrRNA communities profiling.

It is a nice pice of work which deserves publication. Although I still reserve my concern about unique PCRs to produce representative amplicons.